mathematical modelling/applied mathematics

COVID-19, queues, shopping, unsafe interactions, viral exposure

**Author for correspondence:**
C. Budd
e-mail: C.J.Budd@bath.ac.uk

# Assessing risk in the retail environment during the COVID-19 pandemic

C. Budd[1], K. Calvert[2,6], S. Johnson[3,5] and S. O. Tickle[4,6]

[1]School of Mathematical Sciences, University of Bath, Bath BA2 7AY, UK
[2]Department of Mathematics, University of Manchester, M13 9PR Manchester, UK
[3]School of Mathematics, University of Birmingham, Birmingham B15 2TT, UK
[4]School of Mathematics, University of Bristol, Bristol BS8 1UG, UK
[5]Alan Turing Institute, London NW1 2DB, UK
[6]Heilbronn Institute for Mathematical Research, University of Bristol, Bristol BS8 1UG, UK

 CB, 0000-0003-4536-1662; KC, 0000-0001-9661-5939; SJ, 0000-0002-8648-1735; SOT, 0000-0002-9229-600X

The COVID-19 pandemic has caused unprecedented disruption, particularly in retail. Where essential demand cannot be fulfilled online, or where more stringent measures have been relaxed, customers must visit shop premises in person. This naturally gives rise to some risk of susceptible individuals (customers or staff) becoming infected. It is essential to minimize this risk as far as possible while retaining economic viability of the shop. We therefore explore and compare the spread of COVID-19 in different shopping situations involving person-to-person interactions: (i) free-flowing, unstructured shopping; (ii) structured shopping (e.g. a queue). We examine which of (i) or (ii) may be preferable for minimizing the spread of COVID-19 in a given shop, subject to constraints such as the geometry of the shop; compliance of the population to local guidelines; and additional safety measures which may be available to the organizers of the shop. We derive a series of conclusions, such as unidirectional free movement being preferable to bidirectional shopping, and that the number of servers should be maximized as long as they can be well protected from infection.

## 1. Introduction

Many of us do our shopping for food, drink and other essential items, in a supermarket or at a takeaway. During the COVID-19, or indeed any other, epidemic this leads to a possible risk of infection. The group of people going shopping is typically *open* in that it is drawn from a diverse population who will come from many different locations and who are, usually, unknown to each other. While this group can be quite large, and diverse, the time spent in contact with each other in such a situation is often relatively short. The question remains as to what is the best way

to organize the dynamics of the shoppers in a supermarket, or takeaway, so as to minimize the overall risk of infection. During the course of the 2020 COVID-19 pandemic various measures have been considered/implemented including directed shopping and the compulsory use of face masks. In this paper, we make a partial assessment of the effectiveness of both of these measures, through the use of mathematical models.

During the epidemic, a typical shopping experience comprises a wait (in a socially distanced queue) by the entrance outside the shop. This queue is then allowed into the shop, typically on a one-in one-out basis. While inside the shop, shoppers are largely free to move as they wish. Finally, on exit, the shoppers form an ordered queue (or queues) to be served. In a takeaway, a similar procedure is involved, although customers typically move straight from the entrance to the serving queue. Unsafe interactions can occur at any of these points. It is natural to want to try and minimize the frequency and duration of any such interactions, while also maintaining the economic viability of the shop. These objectives are not necessarily compatible. This leads to the issue of determining the optimal way of organizing both the 'free-form' shopping (with reasonable constraints consistent with modelling the shopping experience) and also the queue (or queues) being served. This is the 'managing the crowd' principle outlined in [1] and is affected by both the internal geometry of the shop and also by the way the crowd is directed around this geometry. For example, should the crowd move 'randomly' in a self-organized fashion, or should it in some way be 'directed' as an ordered queue throughout the shop. Similarly, in the case of the checkout queue, the safety of the customers in the queue will be affected by the number, and level of protection, of the servers and of each other. These considerations must also be balanced against the risk to the servers themselves.

In this paper, we address some of these issues by constructing a mathematical model of the above shopping process looking at both the movement in the shop and also in the queues. This model helps to determine the total viral dose experienced by an average shopper. It is based on certain simple assumptions of the way that the virus spreads within the shop and between people, and also of the way that the people move within the store as they make their shopping choices. We emphasize that these conclusions are obtained by the use of mathematical models based on certain assumptions and caveats which we describe alongside the models. At this stage, we have not considered any actual data for the COVID-19 case (although we do draw in places on data for other diseases), and the conclusions are the results only of the simulations and mathematical arguments. However, we show through the modelling experiments, and looking at the level of uncertainty in the model predictions, that the conclusions are reasonably robust to changes in the assumptions themselves. We emphasize that these studies are in some senses preliminary, and we hope that they will lead to further, data-driven investigations.

Our approach comprises an agent-based (ABM) social force model of the crowd (of varying density) within the shop acting as 'typical shoppers' [2], combined with a queuing model of the checkout itself, and models for the viral spread and the impact of PPE. Other approaches to studying the issues associated with the retail environment have also been considered by the Royal Society RAMP initiative. In particular, we compare the results of this approach with that considered in [3], which examines several possibilities for managing the crowd within a supermarket environment, including unidirectional aisles and enforced capacity in popular areas of the shop, and determines the viral dose as a function of the arrival rate of the shoppers. See also the use of Poisson process model for calculating the spread of COVID-19 in the retail environment [4].

In §2 of this paper, we consider an agent-based model for free-flowing shoppers in the supermarket, with a probabilistic model for the way in which they move as they choose their goods. Such a model assumes that there are many shoppers who interact with each other over a short period of time, a proportion of whom may be infected. During such interactions the shoppers accumulate a likely *viral dose* which is then linked to their risk of infection. The analysis in §2 comprises the use of a social force model of the crowd making certain assumptions of their mode of shopping, combined with a model for the transport of the virus from one crowd member to another.

In §3, we move the shoppers to the queue, or queues, at the exit of the shop (which they may move to directly if the shop is a takeaway) and use queuing theory to assess the risk of infection in this (as it turns out quite dangerous) situation. A clear problem in this phase of the shopping is that the queue may be slow moving, and this significantly increases the risk to both the shoppers and the servers. In the modelling of the queue, we consider the relative risk to both shoppers and servers, and rigorously study how it changes with the dynamics of the queue, the number of queues/servers, and the level of protection given to both the servers and the shoppers. In particular, we model the wearing of masks by lowering the probability of an infection spread in an unsafe interaction with an infected person.

Applying these mathematical models leads to a number of tentative conclusions which, as described above, are fairly robust to the modelling assumptions that we have made.

## 1.1. Moving around the shop

A first conclusion from modelling the movement of the crowd in §2 is that minimizing the duration of interactions can be more effective than focusing only on the distance between people. (This conclusion depends only weakly on the precise model used for the aerosol transmission.) Thus a 2 m separation may be quite dangerous if the people interacting spend a long time at this distance, whereas a separation of less than 2 m may be safer if the interaction is shorter.

We find that bidirectional shopping leads to a higher number of viral particles inhaled. However, this structure also makes a significant difference to the efficiency of shopping. Unidirectional shopping led to the lowest viral exposure, as long as shoppers went against the direction if they forgot an item. Lower efficiency of shopping leads to shoppers spending longer in store and hence to higher viral doses. Unidirectional shopping, with the assumption that shoppers did break the directional rules if they needed, was the most efficient and led to the lowest exposure in every situation. However, the difference gained or lost by different shopping mechanics was dwarfed by the effect of differing aisle widths. Shops with 2 m aisle widths led to up to three times more viral exposure than the same system in a 3 m or 4 m wide aisle. These conclusions follow the reasonable assumptions that people act in response to social and physical forces and that the viral particles from an individual disperse with isotropic diffusion (proportional to $1/r^2$ with distance $r$ from an individual). This is not known about COVID-19 particles; however, our conclusions are robust to changing the viral model to being proportional to $a/r + (b/r^2) + c/r^3$ for any positive constants $a$, $b$, $c$.

This result complements previous work which has found efficiency to be the main disadvantage to organized shopping [3], and suggests that venues which have already implemented such measures might reconsider their policy if it is leading to significantly longer shopping experiences. It would also be prudent to analyse shop layout and flow with a view to minimizing the time spent shopping. Furthermore, venues which have the capacity to widen aisles should consider implementing this as of utmost importance in lowering risk.

## 1.2. Organizing the exit queue

In §3, we draw the following tentative conclusions for organizing the exit queue(s): (i) unsafe interactions should be kept to a minimum, (ii) protective mask wearing should be maximized, and mandatory for the queue servers, (iii) mandatory extra protection should be provided for the servers, (iv) the number of servers should be maximized (under the constraint of keeping them safe from each other), and (v) wherever possible, customers should be organized into separate, non-interacting queues, each with a single server, rather than a single queue serviced by a number of servers. These conclusions follow from the reasonable assumption that COVID-19 spreads primarily through unsafe interactions which occur between pairs of people, with these unsafe interactions occurring with greater frequency in situations in which, for example, customers are closer together on average. We assume that a given queuing system will have some time-invariant *unsafe interaction rate* for any pair of people who can come into contact in the exit queue. Note that, in practice, this rate is unlikely to be uniform between different pairs; for instance, certain customers may be more observant of social distancing rules than others.

Additionally, we assume that, when an unsafe interaction occurs between an infected and an uninfected person, the virus spreads with some person-invariant probability. Again, in reality, it is likely that there exist 'super-spreaders' of the virus, for whom the probability of transmission is greater, given an unsafe interaction. Transmission probabilities are also known to be affected by, for instance, the length of time since infection.

Central to all of these results is the model of the safety effect of wearing masks. This effect is assumed to be asymmetric: in an unsafe interaction between an uninfected and infected person in which both wear masks, more of the benefit of the mask wearing is believed to come from the mask worn by the infected person, see, for instance, [5]. For this reason, without masks, servers have a potentially increased chance of becoming super-spreaders of the virus within an open group of shoppers. This principle is also seen in the novel theory developed in §3, in which we examine the spread of infection in two queuing systems: (i) multiple queues with a single server each (as in a supermarket); and (ii) a single queue with multiple servers (as in a coffee takeaway shop or a self-service queue in a supermarket).

The conclusion that wearing masks is important is of course of no surprise, indeed mask wearing is now compulsory in UK shops. However, we hope that the reasoning behind this observation will help guide decisions on the future use of masks as the current crises starts to ease.

# 2. Many-to-one interactions in a crowded supermarket

In this section, we consider a two-dimensional model for supermarket shopping which represents the shoppers moving in a crowd as small circles (which are in turn cross-sections of cylinders). The central assumption of this section is that there is a large number, $N$, of shoppers who come from an originally well-mixed and open population, who are in close proximity with each other for a relatively short time period.

## 2.1. A particle model of a supermarket crowd

To simulate, and then to analyse, the crowd of shoppers in a supermarket, we consider a particle model to represent each person in a two-dimensional domain representing a typical supermarket, as illustrated in figure 2. The shoppers will then act as agents, moving around the store according to certain rules governing the way that they are likely to shop, and will come into contact with other shoppers as they do so. This domain can be thought of as a supermarket with constraints such as walls and aisles. We set out to compare the behaviour of the crowd in different shopping structures of the store, and how these structures may influence viral exposure and shopping progress.

In the model, each shopper, indexed by $\alpha$, is considered to be a separate particle with radius $r_\alpha$ centred at position $\mathbf{x}_\alpha$, a velocity $\mathbf{v}_\alpha$ and an acceleration $\mathbf{a}_\alpha$. The positions and velocities of the shoppers then evolve due to the acceleration of the public actors. Such motion is governed using Newton's second law of motion, $\mathbf{f}_\alpha = m_\alpha \mathbf{a}_\alpha$, where $\mathbf{f}_\alpha$ is the force on the public actor (a combination of social force, intelligent intent and geometrical constraint), $m_\alpha$ is the mass and $\mathbf{a}_\alpha$ is the acceleration of the public actor. The system of ODEs that governs the position and velocity is then

$$\frac{\mathrm{d}}{\mathrm{d}t}\mathbf{x}_\alpha = \mathbf{v}_\alpha \quad \text{and} \quad \frac{\mathrm{d}}{\mathrm{d}t}\mathbf{v}_\alpha = \frac{1}{m_\alpha}\mathbf{f}_\alpha,$$

For simplicity, we will absorb $m_\alpha$ into $f_\alpha$. This makes $f_\alpha$ an acceleration not a force; however, we will still refer to $f_\alpha$ as a social force. We model each shopper as a cross-section of a cylinder of radius $r_\alpha$. For our model, we will consider the shoppers to be initially distributed randomly throughout the supermarket. The forces acting on each shopper are then dependent on their surroundings, the nearby shoppers and their intent and mode of shopping. Following [2], we consider four forces that act on the shoppers: a strong repulsion force from the supermarket walls, $\mathbf{f}^{\text{walls}}$; a repulsion force, $\mathbf{f}^{\text{repel}}$, representing social distancing; and an attraction force $\mathbf{f}_\alpha$ representing a shopper's intent to buy a particular item. It is the force $\mathbf{f}_\alpha$ which is most dependent on the individual concerned, and the hardest to model. The force on the shopper $\alpha$, is then expressed as a sum of all the forces discussed, giving us

$$\mathbf{f}_\alpha = \mathbf{f}_\alpha^{\text{walls}} + \mathbf{f}_\alpha^{\text{repel}} + \mathbf{f}_\alpha^{\text{attract}}.$$

The motion of the crowd arising from these forces then depends upon the precise description of each individual force. We now discuss these in detail. Each force is governed by a set of parameters which we then choose informed by established work in agent-based models. However, not every shopper is the same. To avoid homogeneity, every parameter for each shopper is chosen on a normal distribution with a mean informed by established theory and a standard deviation of one-quarter of the mean. In the following descriptions, we give mean values for the parameters.

### 2.1.1. Forces acting on shoppers

*Wall forces*: The wall repulsion force, $\mathbf{f}^{\text{walls}}$, is a force felt by the shopper $\alpha$ from the closest point of the wall $\mathbf{x}_w$ provided that the distance to that point in the wall $d_w = |\mathbf{x}_w - \mathbf{x}_\alpha|$, is smaller than a chosen threshold distance. The resulting force is then given by

$$\mathbf{f}^{\text{walls}} = -\left(f_{\max}^{\text{obstacle}}\frac{1}{1+(d_w/r_\alpha)^p} + g_{\max}^{\text{obstacle}}\exp\left(\frac{d_w}{\sigma_{\text{wall}}}\right)\right)\frac{\mathbf{x}_\alpha - \mathbf{x}_w}{d_w}. \tag{2.1}$$

The first term in the brackets in (2.1) is taken from [2]. This is a mid-range force and models the *desire* of people to not be too close to walls. The second term taken from [6] is the force adjacent to the wall/aisle and models a shopper's inability to move through the walls. In this expression, the constant $f_{max}^{obstacle}$ is the maximum mid-range force, $g_{max}^{obstacle}$ and $\sigma_{wall}$ control the short-range force. Following the recommendations of [7] for parameters $p = 2$ and $f_{max}^{obstacle}$ as around 2 times the maximal acceleration we define $f_{max}^{obstacle}$ to be 4 m s$^{-2}$. We ran experiments to define the values of $g_{max}^{obstacle}$ and $\sigma_{wall}$ such that shoppers used the most amount of space when required but did not pass through the wall. We set $g_{max}^{obstacle} = 1000$ m s$^{-2}$ and $\sigma_{wall} = 0.01$ m and define the threshold distance at 1 m.

*Forces between the shoppers*. The repelling force from one shopper is the social force induced by a territorial sphere, when a pedestrian becomes increasingly uncomfortable the closer they get to a stranger. Following [8], we define the social repulsion forces by

$$\mathbf{f}_{\alpha}^{repel} = -\mathbf{u} V_{\alpha\beta}[b(\mathbf{x}_{\alpha} - \mathbf{x}_{\beta})]. \tag{2.2}$$

In this expression, $\mathbf{u}$, $V_{\alpha\beta}$, $b$ are defined as

$$\mathbf{u} = \frac{\mathbf{x}_{\beta} - \mathbf{x}_{\alpha}}{|\mathbf{x}_{\beta} - \mathbf{x}_{\alpha}|}, \quad V_{\alpha\beta}(b) = V_{\alpha\beta}^0 \exp\left(\frac{-b}{\phi}\right),$$

and

$$2b(\mathbf{x}_{\alpha} - \mathbf{x}_{\beta}) = ((|\mathbf{x}_{\alpha} - \mathbf{x}_{\beta}| + |\mathbf{x}_{\alpha} - \mathbf{x}_{\beta} - \Delta_t^{step}\mathbf{v}_{\beta}|)^2 - (\Delta_t^{step}|\mathbf{v}_{\beta}|)^2)^{1/2}.$$

The expression $b$ is not symmetrical about the shopper $\alpha$, it accounts for pedestrians requiring space for their next step and other pedestrians allowing that space.

Again following [8], we define $V_{\alpha\beta}^0 = 2.1$ m s$^{-2}$, $\phi = 0.3$ m, $\Delta_t^{step} = 2$ s.

*Attractive forces modelling the intelligent intent of the shopper*: Following [2], each shopper has a desired velocity $\mathbf{v}_{\alpha}^{desire}$, which in the context of this paper will model the way in which the shopper will proceed with doing their shopping. The force compelling them to travel at this velocity is

$$\mathbf{f}_{\alpha}^{attract} = \frac{1}{\tau_{\alpha}}(\mathbf{v}_{\alpha}^{desired} - \mathbf{v}_{\alpha}),$$

where $\mathbf{v}_{\alpha}$ is the shopper's current velocity and $\tau_{\alpha} = 1$ s. Calculating the (changing) desired velocity of a shopper $\mathbf{v}_{\alpha}^{desired}$ as they go about their shop is a very subtle part of this modelling procedure, and the most open to the assumptions made on the way that people shop. A shopper entering a supermarket has as a main goal, the desire to pick up the products on their list. Usually, a shopper does not do this randomly, nor do they usually do this with exact precision. As a consequence, while their passage around the store is not completely random, it is also usually sub-optimal. As a model for this behaviour, we randomly generate a number of points in the store domain for each shopper. These points then become their *shopping list* of desired items. In a perfect world, each shopper would sort their list entirely and therefore only have to travel around the aisles in the store in one direction. We model this scenario by initially fully sorting each shoppers list by considering the location of the items in the store along an optimal path. However, in reality shoppers might forget something, or not be infallible in their organization, or simply not know in advance where the item which they want to buy is in the store. (In the authors' experience this is the rule rather than the exception!) Hence, they will often have to double back during their passage around the store. This motion can then be represented by a partially sorted list. We implement this by taking the fully sorted list and applying a small number of random permutations to it to give a partially sorted list. The shopper will then move around the store going from one item in this partially sorted list to the next. Let $\mathbf{x}_{desired}$ represent the desired point of a shopper $\alpha$ with desired speed $s_{\alpha}$ we define the desired velocity

$$\mathbf{v}_{\alpha}^{desired} = s_{\alpha} \frac{\mathbf{x}_{\alpha} - \mathbf{x}_{desired}}{|\mathbf{x}_{\alpha} - \mathbf{x}_{desired}|}.$$

We take $s_{\alpha}$ to be from a normal distribution with mean 1.4 m s$^{-1}$, this follows data acquired by [9] from measurements in shopping centres.

*Supermarket structure, unidirectional, bidirectional, strict-unidirectional*. We have designed our model to implement three different supermarket structures. The first structure is a one-way system. Every shopper is told to go in one direction around the shop. However, in this system shoppers do go back on themselves. The second structure is a two-way system where shoppers are allowed to travel either way around the supermarket. The final structure, a strict one-way structure. In this system, if a

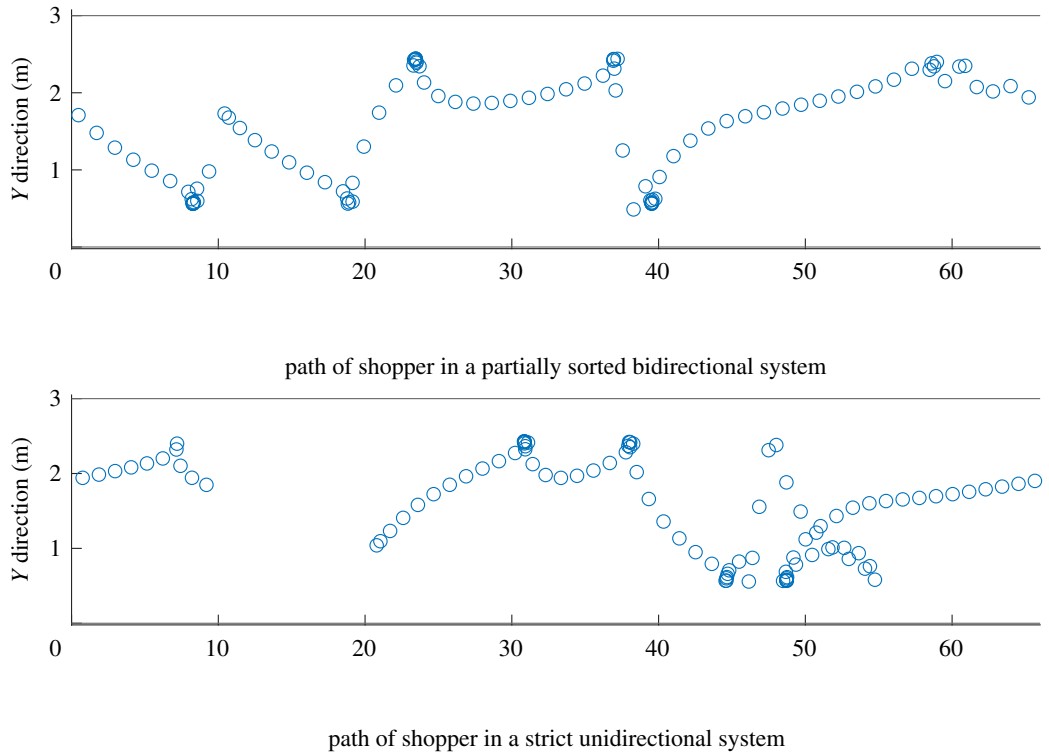

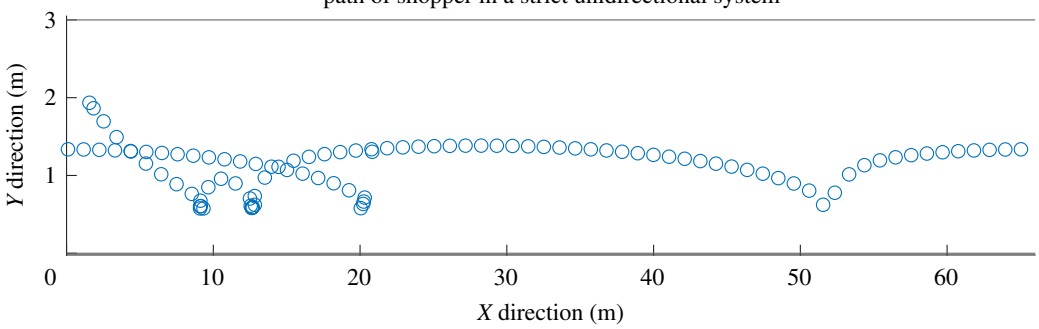

**Figure 1.** Plots of the path of a single shopper moving around a looped aisle with differing shopping structures: sorted, partially sorted bidirectional, partially sorted strict unidirectional.

shopper had to turn back more than 1 m they do not break the rules and they go all the way around the loop again. We believe this is less realistic. Note that if shoppers' lists were fully sorted both one-way systems would be identical. In figure 1, we display the path of an individual in a 3 m by 66 m empty aisle under different supermarket structures.

We presume that each shopper is guided by their partially ordered list as follows. They are initially attracted to the position of the first item on the list. When they get within 1 m of this item then at every time step they pick up this item with probability $p$. This models the empirical observation that shoppers may take some time to decide which precise item they want to buy. Once they pick up that item they are then attracted to the second item in their list and this continues until the list is complete. In our model, we monitor the progress of the shoppers. This is defined to be the number of items a shopper has picked up in the time-frame during which the model is run.

### 2.1.2. Viral exposure

We next consider calculating the viral dose of customers in the supermarket as they move through the store.

In our model, we will consider that one (random) individual is deemed to be infected. The infected individual is asymptomatic. This information originates from experiments done by [10] which agree with previous experiments [11–13].

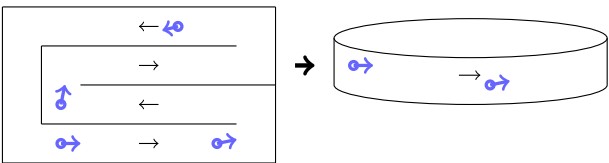

**Figure 2.** Modelling an aisle as a corridor loop.

### 2.1.3. The density of the viral particles

We next make the assumption that the viral density is proportional to $r^{-2}$, where $r$ is the distance from the infected individual. This assumes that the viral particles disperse uniformly with distance; that is, there is an equal number of viral particles in every 1 m annulus around the infected individual. A better understanding of the mechanics of the COVID-19 particle transmission in the air will naturally lead to improvements on this model. However, later in this paper, we will consider other rules for the decay of the viral density with distance and will show that the conclusions from the model are fairly robust to the precise details of the particle dispersion. Our model of the density of the viral particles in the air will thus be initially

$$\rho(t, r) = \frac{\Lambda}{r^2} \quad \text{particles m}^{-3}, \tag{2.3}$$

for a suitably chosen constant $\Lambda$. We will choose $\Lambda = 10^3$ so that $\rho(0.1\,\text{m}) = 10^5$ particles m$^{-3}$. This matches the experimental measurements of viral particles under normal breathing [10–13].

*Exposure of individuals to the viral particles.* Let $V^{\text{inhaled}}$ be the proportion of the surrounding air inhaled by an individual. We model the viral dose $\sigma_\alpha$ inhaled by a healthy shopper $\alpha$ at a distance $r = |\mathbf{x}_\alpha - \mathbf{x}_{\text{infected}}|$ from an infected individual located at the position $\mathbf{x}_{\text{infected}}$ as

$$\frac{\mathrm{d}}{\mathrm{d}t}\sigma_\alpha = \Lambda \frac{V^{\text{inhaled}}}{\left|\mathbf{x}_\alpha - \mathbf{x}_{\text{infected}}\right|^2}.$$

The average lung capacity is 6 l [14] and the average breath rate is 12–20 min$^{-1}$ [15]. If we assume that an individual breathes from a 1 m$^3$ volume every 4 s (a rate of 15 min$^{-1}$), then we can estimate $V^{\text{inhaled}}$ as $6 \times 10^{-3} \times 1/4 = 1.5 \times 10^{-3}$ m$^3$. For every individual, we define $V^{\text{inhaled}}_\alpha$ to be a value taken from a normal distribution with mean $V^{\text{inhaled}}$ and standard deviation $0.25 V^{\text{inhaled}}$.

This subsection describes what is only a rough estimation of the behaviour of the viral particles. The full aerodynamic motion of viral particles is hard to simulate, we are not able to include that in our simplified model. We have, however, run the exact same visualizations with viral density proportional to $1/r$, $1/r^2$ and $1/r^3$. Although the exact results were different for the differing exponents, we found that the comparisons of viral exposure per item were still valid and robust under the change of exponent. One could view the viral density as a probability density of viral particles. If the particles decay slowly or currents spread them further, then an exponent larger than −2 may be required, while if particles drop to the ground quickly then perhaps an exponent smaller than −2 is more appropriate. This is expanded in §2 of the electronic supplementary material.

Every parameter except $\Lambda$, $\sigma_{\text{wall}}$ and $g^{\text{obstacle}}_{\text{max}}$ were set from recommendations of well-established agent-based models [2,6,7]. We ran experiments to set $\sigma_{\text{wall}}$ and $g^{\text{obstacle}}_{\text{max}}$. Slightly varying these values had the effect of narrowing the domain but did not change the results very much. Our aim is to make comparisons between different shopping mechanics, the value of $\Lambda$ does not alter the comparisons that we make. We will not be using the actual values of viral exposure in our conclusions. The domain we use for the numerical experiments is a looped aisle which models a structured aisle-based system as shown in figure 2. This is a simplification of a supermarket. The results stated here apply to a simple loop, further work would need to be conducted to rigorously conclude this applies to more complicated domains.

## 2.2. Results

We now consider combining the above models of the crowd dynamics, and of the viral exposure, to determine the viral dose of a typical shopper moving in the crowd. We compare the viral exposure encountered in the different structured shopping environments discussed in the section on supermarket structure in §2.1.1.

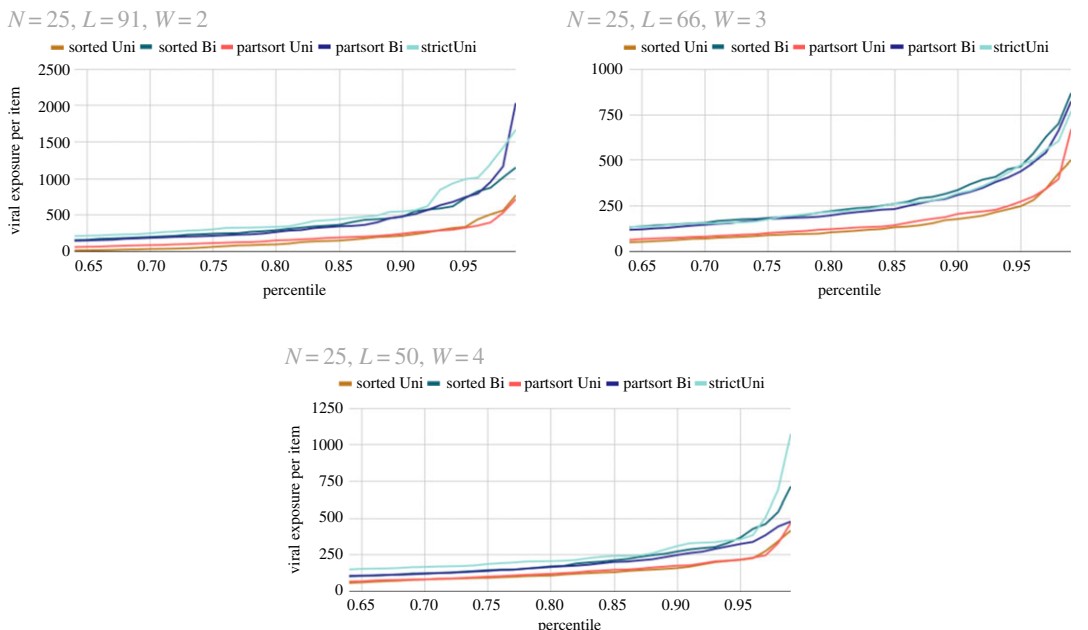

**Figure 3.** Plots of viral exposure per item for differing shopping structures and population $N = 25$ in an aisle widths of 2, 3 and 4 m. We refer the reader to figure 1 of the electronic supplementary material for plots with different values of $N$.

### 2.2.1. Numerical experiments

For our numerical experiments, we fix the looped shopping aisle to have an area of 200 m². We compare three different aisle widths: 4, 3 and 2 m, three different populations numbers: 7, 15 and 25, and five different shopping list structures: sorted unidirectional, sorted bidirectional, partially sorted unidirectional, partially sorted bidirectional and partially sorted strict unidirectional. We then ran the crowd simulation model for a series of 20 visualizations at each crowd population $N$, aisle width $W$ and supermarket structure. The simulation modelled a typical shopping experience for $T = 15$ min. The model described is implemented on MATLAB [16] using the ODE solver ode45 [17]. We recorded the viral dose $\sigma_\alpha$ for each individual $\alpha$, the number of items picked up items$_\alpha$, and the viral exposure per product $\sigma_\alpha$/items$_\alpha$.

### 2.2.2. Conclusions from the numerical experiments

For *larger* aisle widths ($W \geq 3$ m figure 3), the different shopping mechanics made little effect on the total viral dose $\sigma_\alpha$ for a given density and geometry. However, the mechanics made a substantial difference to the progress of shoppers items$_\alpha$. A strict unidirectional system was the least efficient for shoppers while the fully sorted unidirectional system was most efficient, although perhaps a little unrealistic. This suggests that a supermarket with larger aisle widths should choose rules and regulations that increase efficiency. For *narrow* aisle widths ($W = 2$ m, figure 3), bidirectional shopping led to higher viral doses and less efficient shopping, this was due to shoppers taking more time to get past each other. Strict unidirectional shopping was also poor because it forces more shoppers to pass each other.

In every simulation when shoppers travelled in one direction and were happy to turn back if necessary the viral dose per item was lowest. However, strict unidirectional shopping where shoppers would not turn back if they had missed an item often led to the highest values of viral dose per item. When a shopper did not turn back they had to do a full loop to retrieve an incorrectly ordered item, this meant they passed or even got stuck behind many shoppers.

Aisle width made a huge impact on viral exposure. Shoppers in a 4 or 3 m wide aisle were exposed to half as many viral particles as shoppers in a 2 m aisle. For aisle widths of 2 m, it is more important that a unidirectional system is implemented. In bidirectional models of narrow aisles, shoppers struggled to move past each other, both increasing viral dose and severely decreasing the efficiency of the shopping experience. It would be prudent for supermarkets to evaluate their shoppers experience. If a supermarket can widen their aisles by 50 per cent they could potentially decrease viral doses experience by a factor of 10. If this is not possible, the driving factor of exposure in a supermarket

appears to be time spent in the supermarket. Work should be done to make shopping as efficient as possible. This could be done by implementing a one-way system as seen in many current supermarkets; however, it should somehow be communicated or emphasized that breaking this one-way system in the name of efficiency is good. Also supermarkets would benefit from arranging their shops in a way such that shoppers spend the least amount of time inside. These conclusions agree with [3] that the driving factor of viral dose was time spent in the supermarket.

## 2.3. Walking past an infected individual with various decay laws for the droplet density

The model described above also allows for simple calculations about the viral dose associated with different possible trajectories of both the infected and the susceptible shoppers. For example, we can ask whether it is better to walk past an infected individual, coming quite close briefly, or to remain for a longer period of time at a safer distance. This question may be relevant when deciding whether to allow shoppers to walk about freely, or rather to organize them into some kind of queue or structured flow. We now consider this, and also look at the robustness of our conclusions to changes in model concerning the rate of spread of the virus droplets.

Consider a susceptible individual who walks in a straight line past an infectious person at a relative velocity $v$, passing them at a minimum distance $\delta$. We now compare this situation with that in which another uninfected individual remains at a constant distance $D$ from the infectious individual for a time $T$. The viral dose $\sigma$ received by the moving individual is

$$\sigma_m = \int_{t=-\infty}^{\infty} \rho(r, t)\, \mathrm{d}t = \int_{t=-\infty}^{\infty} \frac{\Lambda}{v^2 t^2 + \delta^2}\, \mathrm{d}t = \frac{\pi \Lambda}{\delta v}, \tag{2.4}$$

where $r(t) = \sqrt{v^2 t^2 + \delta^2}$ is the distance between the infected and susceptible individuals at time $t$, and $\rho(r, t)$ is given by equation (2.3). For the static individual the viral dose will be

$$\sigma_s = \frac{\Lambda T}{D^2}. \tag{2.5}$$

Whether it is preferable to be static or moving depends on the ratio $\sigma_m/\sigma_s$ (regardless of the probability of infection given a viral dose, which we analyse below). There will thus be a critical shortest distance $\delta_c$ such that it is safer to walk past than remain at a distance $D$ from an infectious individual

$$\delta_c = \frac{\pi D^2}{vT}. \tag{2.6}$$

Note that this expression does not depend on uncertain quantities such as $\Lambda$.

In the above and in the previous subsections, we have considered the density of virus in the air, $\rho$, to decay with the square of the distance from the viral source. This corresponds to making the assumption that the particles move primarily by isotropic diffusion in three dimensions. To test the robustness of the conclusions from this assumption, we now generalize equation (2.3) to account for either a shorter or a longer range decay, to take the more general form

$$\rho(r) = \frac{\Lambda}{r^\gamma}, \tag{2.7}$$

with $\gamma$ a constant. If the virus is only carried in relatively large droplets, which tend to fall to the ground, then this would translate into taking a value of $\gamma > 2$. However, there is also evidence that these droplets can be transported on convection currents, such as wind, air-conditioning, coughing and sneezing. Such a transmission method would then decrease the value of $\gamma$. It is also thought that the virus can be carried by aerosols, such as small droplets created by ventilation in hospitals, or other particles in polluted environments [18]. These would diffuse further than larger droplets, making $\gamma$ closer to 2. The value of $\gamma$ will also depend on conditions such as temperature and humidity. For instance, a drier environment will result in smaller droplets which can diffuse further.

For values of $\gamma \geq 2$, the viral doses for the static and moving individuals are, respectively, given by the expressions

$$\sigma_s = \frac{\Lambda T}{D^\gamma} \tag{2.8}$$

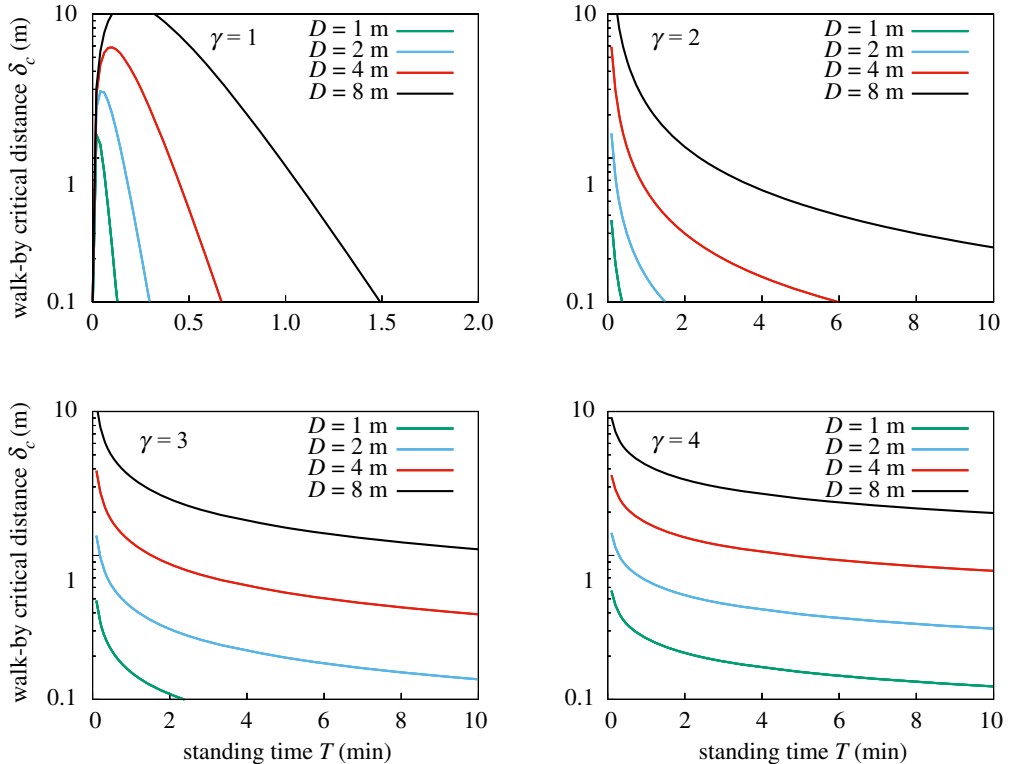

**Figure 4.** Critical distance, $\delta_c$, against exposure time, $T$, for static individual distances $D = 1$, 2, 3 and 4 m, as given by equations (2.10) and (2.12). Panels are for $\gamma = 1$, 2, 3 and 4, where $\rho \propto r^{-\gamma}$.

**Table 1.** Several values of the coefficient $\beta_\gamma$.

| $\gamma$ | 2 | 3 | 4 | 5 | 6 | 7 | 8 | 9 | 10 |
|---|---|---|---|---|---|---|---|---|---|
| $\beta_\gamma$ | $\pi$ | 2 | $\frac{\pi}{2}$ | $\frac{4}{3}$ | $\frac{3\pi}{8}$ | $\frac{16}{15}$ | $\frac{5\pi}{16}$ | $\frac{32}{35}$ | $\frac{35\pi}{128}$ |

and

$$\sigma_m = \frac{\beta_\gamma \Lambda}{v \delta^{\gamma-1}}, \tag{2.9}$$

where $\beta_\gamma$ is a coefficient. The values of $\beta_\gamma$ for integer $\gamma \in [2, 10]$ are given in table 1.

The critical distance is therefore

$$\delta_c = \left(\frac{\beta_\gamma D^\gamma}{vT}\right)^{1/(\gamma-1)}. \tag{2.10}$$

In the case that $\gamma = 1$, the expression in (2.4) diverges. We therefore consider that the total duration of the walk is $T$ (instead of infinity). In this case, we obtain

$$\sigma_m(\gamma = 1) = \frac{2\Lambda}{v} \ln\left(\frac{2vT}{\delta}\right), \tag{2.11}$$

which gives a critical distance

$$\delta_c(\gamma = 1) = 2vT \, \exp\left(-\frac{vT}{2D}\right). \tag{2.12}$$

Note that the only uncertain quantity appearing in these expressions for the critical distance, $\delta_c$, is the decay exponent, $\gamma$. Figure 4 shows $\delta_c$ against exposure time $T$, for different values of the static individual's distance $D$ and the exponent $\gamma$. The velocity is taken to be $v = 1.4$ m s$^{-1}$, which is a normal walking speed.

As we can see, $\delta_c$ is quite low for $T$ on the order of a few minutes and $D$ of metres. For example, in the $\gamma = 2$ case, it would be safer to walk past an infectious individual, passing at just $\delta = 10$ cm, than to remain for $T = 2$ min at a distance of $D = 2$ m. Even if $\gamma = 4$, it is safer to walk past at 50 cm than to spend 5 min at 2 m.

# 3. Queuing and structured shopping

## 3.1. Overview

In this section, we examine the spread of COVID-19 in a structured queue at the exit of the shop. We consider two different models for the queuing system, and two different models for infection within the queue, with various levels of personal protection. For the queue, we examine

  (i) $k$ queues in parallel, each with a single server (as in a supermarket); and
  (ii) one queue with $k$ servers (as in a takeaway shop).

For the possible infections in the queue, we examine

(a) *all-to-all interactions*—any two individuals in the same queue have the potential to directly spread infection to one another; and
(b) *nearest neighbour only interactions*—COVID-19 can only directly spread between *adjacent* customers in the same queue.

## 3.2. Modelling assumptions

We now briefly discuss some important modelling assumptions made regarding the shop and the transmission behaviour of COVID-19.

### 3.2.1. Shop assumptions

*Capacity*: In setting (ii), we have a single queue with capacity $C$. This capacity includes anyone who is currently being served by one of the servers, so trivially we require $C \geq k$. Note we assume that entry is controlled so that the queue never exceeds length $C$. In setting (i), we have $k$ queues each with capacity $C'$. For each queue, this capacity includes the customer being served. We may think about this queue capacity in the supermarket setting as a hard policy adopted by shop managers to prevent queuing into customer browsing space, so $C'$ would typically be on the order of 2 or 3.

*Unsafe interactions*: For this section, we examine only the spread of COVID-19 directly from person to person. We ignore infections which may arise from, for example, the sharing of contaminated surfaces. In addition to possible customer-to-customer interactions—which we assume occur either according to regime (a) or (b)—a server and the customer they are currently serving may also have an unsafe interaction. Finally, we assume that servers on duty at the same time can have unsafe interactions with one another. We take the arrival rate into the queue as $\mu(t)$ hour$^{-1}$ at time $t$, and the service rate of an individual server as $\lambda$ hour$^{-1}$, assuming that they are always busy. Meanwhile, the rate at which an unsafe interaction occurs depends on who is affected by the interaction. We take

— $\xi_c$ hour$^{-1}$, the unsafe interaction rate between any valid pair of customers in the queue system;
— $\xi_S$ hour$^{-1}$, the unsafe interaction rate between any two servers on duty in the queue system;
— $\xi_{SC}$ hour$^{-1}$, the unsafe interaction rate between any server and the customer they are currently serving.

*Masks and additional safety*: Our final shop-based assumption is that it is possible to mandate that all servers wear masks, and that some form of additional shielding of servers from the shoppers (e.g. a screen at the counter) is possible.

### 3.2.2. COVID-19 assumptions

*Unsafe interactions*: As discussed above, an unsafe interaction is taken to be central to the transmission of COVID-19. When an unsafe interaction occurs between an infected and an uninfected person, we say that the probability of infection spreading is $p$.

*Mask protection*: We assume that the probability of infection transmission will be reduced if one or both of the people in an unsafe interaction are wearing masks. This effect will be different depending on whether the infected or the uninfected person is wearing a mask. Formally, if the uninfected person wears a mask, and the infected person does not, the transmission probability is $\alpha_1 p$; if, instead, the uninfected person does not wear a mask, and the infected person does, the transmission probability is $\alpha_2 p$; finally, if both the uninfected and infected people wear masks, the transmission probability is $\alpha_1 \alpha_2 p$.

We remark that the extent of the effectiveness of mask wearing in reducing the spread of COVID-19 is a subject of much ongoing research. A pessimistic viewpoint is possible in the following theory by letting $\alpha_1 = \alpha_2 = 1$, with no substantive change to the remainder of our conclusions. We do not attempt to estimate these two values in this article.

*Additional safety*: As for masks, we assume that the additional safety at the counter provides protection by reducing infection probability. If this safety is present, the probability of transmission becomes $\beta p$. This protection factor is the same regardless of whether the server or the shopper is infected in the interaction. Note that if both masks and extra protection are present, we assume that their effects are independent of transmission probability. For example, the probability of transmission from an infected server to an uninfected customer in which both are wearing masks and extra protection is present is taken as $\alpha_1 \alpha_2 \beta p$.

*Starting assumptions*: We take the initial starting proportion of the population infected by COVID-19 to be $p_0$. We assume that nobody is immune.

## 3.3. Variables of interest

We now discuss the factors which we shall examine in our analysis of a shop queue system over a given period of time. These variables can be divided into two types: we focus on three variables with respect to the *behaviour of the population*

— $\xi_C$ hour$^{-1}$, the unsafe interaction rate between any valid pair of customers in the queue system;
— $\xi_{SC}$ hour$^{-1}$, the unsafe interaction rate between a server and the customer being served; and
— $p_M$, the proportion of the shopping population who wear masks.

Our other variables concern *shop policy decisions* associated with the management of the queuing system

— $\xi_S$ hour$^{-1}$, the unsafe interaction rate between any two servers on the same shift in the queuing system;
— $k$, the number of servers in a shift;
— whether servers are mandated to wear masks or not (we label this with the indicator $\gamma_S$, such that $\gamma_S = 1$ if servers must wear masks, and $\gamma_S = 0$ otherwise); and
— whether additional protection (for example, a small screen) is placed between servers and customers. Again, we label this with an indicator, $\gamma_E$.

## 3.4. Theory

We explore, for both queuing models (i) and (ii), and both infection models (a) and (b), the theoretical relationship between these variables and the number of shoppers who become infected as a result of visiting the shopping queue over a period of time, say $T$ hours of business. We emphasize that all results in this section are subject to the assumptions discussed.

Let $\pi_i$ be the probability that, at equilibrium, there are exactly $i$ people in a queue of type (ii). For $k$ servers and a capacity of $C$, the values of $\pi_i$ are stated in §4 of the electronic supplementary material. Alternatively, e.g. [19].

Meanwhile, establishing the equilibrium probabilities for a queuing system of type (i) is a subject of much recent interest and progress. For example, Dester *et al.* [20] have successfully derived the steady-state probabilities of the system when we allow $C' \to \infty$. To the best of our knowledge, the general steady-state probabilities of this queuing system are still unknown when we insist upon an arbitrary finite capacity in each queue. If we let $\epsilon_{n_1,\ldots,n_k}$ be the steady-state probability that there are $n_i$ people in queue $i$ for $i \in \{1, \ldots, k\}$, then lemma 4.1 in §4 of the electronic supplementary material gives the mechanism for deriving the steady-state probabilities for small $C'$ and $k$.

For convenience in some of the later results, let $\epsilon_j = \sum_{n_k=0}^{C'} \cdots \sum_{n_2=0}^{C'} \epsilon_{j,n_2,\ldots,n_k}$ be the steady-state probability that the first queue has exactly $j$ customers present. Note that by symmetry of the arrivals and services, the choice of the first queue is without loss of generality, with $\sum_{j=0}^{C'} \epsilon_j = 1$.

With these equilibrium probabilities for the length of the queue, we are in a position to find the number of people who become infected in each queuing system and infection model pairing. We begin with infections accrued by shoppers from other shoppers, assuming we start with a large denominator population of potential users of the shop.

**Lemma 3.1.** *In addition to the assumptions of §3.2, assume that any shoppers infected by servers do not spread the infection to any more shoppers before leaving the shop. Let $I(T)$ be the number of shoppers who are infected by other shoppers after a time $T$ from the large denominator population which uses the queue in the shop, and let*

$$q = p_0(1 - p_0)p(1 - p_M + \alpha_1 p_M)(1 - p_M + \alpha_2 p_M).$$

*Then*

$$
\mathbb{E}[I(T)] = \begin{cases}
\xi_C kTq \sum_{j=2}^{C'} \epsilon_j j(j-1) & \text{in case (a)(i)} \\
\xi_C kTq \sum_{j=2}^{C'} 2\epsilon_j(j-1) & \text{in case (b)(i)} \\
\xi_C Tq \sum_{j=2}^{C} \pi_j j(j-1) & \text{in case (a)(ii)} \\
\xi_C Tq \sum_{j=2}^{C} 2\pi_j(j-1) & \text{in case (b)(ii).}
\end{cases}
$$

*Proof.* See §4 of the electronic supplementary material. ∎

We remark that the expected number of newly infected shoppers is linear in both $\xi_C$ and $T$. This is unsurprising, but underlines the importance of appropriate social distancing to minimize unsafe interactions. Additional precautions to reduce unsafe interactions such as minimizing talking indoors could also be taken. See, for example, [21].

Additionally, given that $0 < \alpha_1, \alpha_2 < 1$, we have that the contribution from the terms involving $p_M$ decreases quadratically with increasing $p_M$. For $p_M = 0$, this contribution is trivially 1, indicating no benefit; for $p_M = 1$, the expected number of infected shoppers is discounted by $\alpha_1 \alpha_2$. Note that some recent efforts, such as [22], have suggested that $\alpha_1 \alpha_2$ could be as low as $1/36$ for COVID-19. This emphasizes the possible value of mask wearing, especially when $p_0$ is non-trivial.

We note the importance of reducing the weighted sum of the stationary probabilities in each system in keeping the expected number of new cases as low as possible. This sum is larger if the system is closer to full capacity most of the time. In setting (ii), for a fixed service rate $\mu$, arrival rate $\lambda$, and queue capacity $C$, it is expedient to make the number of servers $k$ as large as possible.

We conclude our commentary on this result by comparing the two different queuing systems under each of the two models for infection spread. Under all-to-all interactions, we note that it is preferable, according to lemma 3.1, in almost all circumstances, to separate customers into $k$ different queues, each staffed by a different server. This is despite the fact that a single queue with $k$ servers results in a faster average service time for a given customer. This heuristic can be seen by comparing the two relevant quantities from lemma 3.1: suppose that the reverse is true, and in fact a single queue leads to fewer customer infections, then we have

$$\xi_C kTq \sum_{j=2}^{C'} \epsilon_j j(j-1) > \xi_C Tq \sum_{j=2}^{kC'} \pi_j j(j-1). \tag{3.1}$$

(Given that we are directly comparing the systems, we assume the same capacity in each case, so that $C = kC'$ in this instance.) To further the 'worst-case' infection spread for the multiple queue system, let us suppose the system is always as busy as possible, i.e. $\epsilon_{C'} = 1$. Therefore, as

$$\sum_{j=2}^{kC'} j(j-1) = \frac{kC'(kC'+1)(kC'-1)}{3},$$

then to satisfy equation (3.1) we need that if $\pi_j > \pi$ for some $\pi \; \forall j \geq 2$, then this $\pi$ must satisfy

$$\pi < \frac{3(C'-1)}{(kC'+1)(kC'-1)}.$$

We therefore see immediately that unless $k \geq 4$ then for a given capacity of $C = kC'$ it is always preferable to split the customers into separate queues which, while slower, are assumed to not to be able to spread infection between one another. Even for $k \geq 4$, the conditions under which a single queue is preferred are strict, with the queue almost always empty, in contrast to the system of $k$ queues which are almost always full. Given that arrival and service rates can vary considerably over the course of trading, under an all-to-all infection assumption, single-server queues should be preferred to a single, multiple-server queue.

We now look at nearest-neighbour only infections, using the same heuristic comparison between the two choices of queue management. Suppose that $k$ separate queues will again lead to more customers becoming infected, so that we have

$$\xi_C k T q \sum_{j=2}^{C'} 2\epsilon_j (j-1) > \xi_C T q \sum_{j=2}^{kC'} 2\pi_j (j-1). \tag{3.2}$$

As per the previous analysis, using equation (3.2), taking $\epsilon_{C'} = 1$ and $\pi_j > \pi \forall j \geq 2$, we attain

$$\pi < \frac{2(C'-1)}{C'(kC'-1)}.$$

This inequality is much less severe than for the all-to-all setting. Therefore, in the nearest-neighbour setting under certain circumstances (for instance, when a single queue leads to much shorter waiting times for the average customer), a single queue may be preferable to multiple queues, particularly if both $C'$ and $C$ are large.

Therefore, our advice from this result is that, if socially distancing customers may be difficult to constantly monitor, or if the layout of the shop precludes an assumption of only neighbours in a queue infecting one another, then multiple queues should be preferred. If, however, the queuing system is in a safer location which guarantees social distancing between customers with low risk of infections between non-neighbours in a queue (for example, a well-marked outdoor queuing area) then the analysis is less clear-cut.

We now examine the number of servers who become infected within a given period of time under each queuing system and infection model.

**Lemma 3.2.** *Suppose $k^*$ is the number of servers who are infected at the time $t = 0$ in queuing system (i). Then, the expected number of newly infected servers in time $T$ is the largest $l \in \mathbb{N}$ such that*

$$\sum_{i=0}^{l-1} \frac{1}{Q(k^* + i, k)} \leq T \left( \xi_{SC} \sum_{n_k=0}^{C'} \cdots \sum_{n_1=0}^{C'} \left( \sum_{i=1}^{k} \mathbb{1}\{n_i > 0\} \right) \epsilon_{n_1,\dots,n_k} + \xi_S \binom{k}{2} \right), \tag{3.3}$$

*where*

$$Q(b, k) = \frac{k-b}{k} p p_0 (1 - \gamma_E + \gamma_E \beta)(1 - \gamma_S + \gamma_S \alpha_1)(1 - p_M + p_M \alpha_2)$$

$$\times \frac{\xi_{SC} \sum_{n_k=0}^{C'} \cdots \sum_{n_1=0}^{C'} \left( \sum_{i=1}^{k} \mathbb{1}\{n_i > 0\} \right) \epsilon_{n_1,\dots,n_k}}{\xi_S \binom{k}{2} + \xi_{SC} \sum_{n_k=0}^{C'} \cdots \sum_{n_1=0}^{C'} \left( \sum_{i=1}^{k} \mathbb{1}\{n_i > 0\} \right) \epsilon_{n_1,\dots,n_k}}$$

*for $b < k$.*

*Proof.* See §4 of the electronic supplementary material. ∎

Note that in §4 of the electronic supplementary material, we give a corollary to this result on the number of customers who become infected by infected servers under a queue system of type (i). (See electronic supplementary material, corollary 4.1.4.)

We now turn to queuing system (ii), and examine the number of infected servers in a single queue with $k$ servers.

**Lemma 3.3.** *Suppose $k^*$ is the number of servers who are infected at the time $t = 0$ in queuing system (ii). Then, the expected number of newly infected servers in time $T$ is the largest $l \in \mathbb{N}$ such that*

$$\sum_{i=0}^{l-1} \frac{1}{P(k^* + i, k)} \leq T \left( \xi_S \binom{k}{2} + \xi_{SC} \left\{ k - \sum_{i=0}^{k-1} (k-i) \pi_i \right\} \right), \tag{3.4}$$

*where* $(\xi_S \binom{k}{2} + \xi_{SC}\{k - \sum_{i=0}^{k-1}(k-i)\pi_i\})/((k-b)p)P(b,k)$ *is equal to*

$$b(1 - \gamma_S + \gamma_S\alpha_1\alpha_2)\xi_S + \frac{p_0}{k}(1 - \gamma_E + \gamma_E\beta)(1 - \gamma_S + \gamma_S\alpha_1)(1 - p_M + p_M\alpha_2)\left\{k - \sum_{i=0}^{k-1}(k-i)\pi_i\right\}\xi_{SC},$$

*for* $b < k$.

*Proof.* See §4 of the electronic supplementary material. ∎

Again, note that the electronic supplementary material contains a corollary to this result on the number of customers who become infected by infected servers. (See electronic supplementary material, corollary 4.1.3.)

Lemmas 3.2 and 3.3 (and electronic supplementary material, corollaries 4.1.4 and 4.1.3) indicate the importance of mandating masks for servers for small $\alpha_1$ and $\alpha_2$, as well as installing additional safety between the servers and the shoppers. Letting $\gamma_S = 1$, we see that both $P(b, k)$ and $Q(b, k)$ are discounted by at least $\alpha_1$ compared to when $\gamma_S = 0$, while fixing $\gamma_E$. Letting $\gamma_E = 1$, we see that both $P(b, k)$ and $Q(b, k)$ are discounted by $\beta$ compared to when $\gamma_E = 0$, while fixing $\gamma_S$. Therefore, if both masks and extra protection are mandated for servers, we can think of the exposure that they get from the shoppers (and each other) as equivalent to that received over a much shorter length of time in a no-protection environment.

We conclude our analyses of these lemmas on interactions involving servers with a heuristic comparison between the two queuing systems. To examine which system leads to fewer servers becoming infected, we take $\xi_{SC} = \xi_C = \xi$, for some $\xi$ for queuing systems (i) and (ii), with $\xi_S = 0$ for queue system (i), and with $\xi_S = \xi$ for queue system (ii). This corresponds to thinking about a traditional supermarket server system in which servers do not interact with one another, versus a coffee shop scenario in which servers may need to interact with one another. We emphasize that scenarios in which customers are serviced by separate queues but servers can still interact are not considered in the following heuristic appraisal of the results above.

Note that if we let

$$Q'(b, k) := Q(b, k)\left(\sum_{n_k=0}^{C'}\cdots\sum_{n_1=0}^{C'}\left(\sum_{i=1}^{k}\mathbb{1}\{n_i > 0\}\right)\epsilon_{n_1,\ldots,n_k}\right)$$

and

$$P'(b, k) := P(b, k)\left(\binom{k}{2} + k - \sum_{i=0}^{k-1}(k-i)\pi_i\right),$$

where $Q(b, k)$ and $P(b, k)$ are as defined in lemmas 3.2 and 3.3, respectively, then queue system (i) leads to a longer time to the infection of the next server for a given $b, k$ if $Q'(b, k) < P'(b, k)$, with (ii) the safer system for servers if the reverse holds. Suppose a single queue in which servers mix leads to a longer infection time, so that $Q'(b, k) > P'(b, k)$. This assumption leads to the inequality

$$\sum_{n_k=0}^{C'}\cdots\sum_{n_1=0}^{C'}\left(\sum_{i=1}^{k}\mathbb{1}\{n_i > 0\}\right)\epsilon_{n_1,\ldots,n_k} - k + \sum_{i=0}^{k-1}(k-i)\pi_i > \frac{kb}{p_0}\frac{1 - \gamma_S + \gamma_S\alpha_2}{(1 - \gamma_E + \gamma_E\beta)(1 - p_M + p_M\alpha_2)}.$$

We make further pessimistic assumptions in favour of the single queue system, analogous to our heuristic discussion of customer infections above. Suppose $\pi_i > \pi > 0$, $\forall i \in \{0, \ldots, k-1\}$ and note that

$$\sum_{n_k=0}^{C'}\cdots\sum_{n_1=0}^{C'}\left(\sum_{i=1}^{k}\mathbb{1}\{n_i > 0\}\right)\epsilon_{n_1,\ldots,n_k} - k \leq 0.$$

If we let the maximum be attained, then note that

$$\pi > \frac{2b}{(k+1)p_0}\frac{1 - \gamma_S + \gamma_S\alpha_2}{(1 - \gamma_E + \gamma_E\beta)(1 - p_M + p_M\alpha_2)},$$

so that we infer that multiple queues should be preferred from the perspective of protecting servers in the following situations in particular

— when the number of already infected servers is suspected to be non-trivial;

— when the proportion of the population currently with COVID-19 is low;

— when the proportion of customers wearing masks is high;

— when extra protection (e.g. a screen) is possible for servers.

We conclude this section with two important additional caveats to these theoretical results. Firstly, we note the importance of assuming a large population and relatively small time $T$. With these assumptions, we can assume that $p_0$ remains fixed for arriving customers throughout. In practice, even for relatively small $T$, it may be the case that some shoppers return multiple times to the same shop, with it being increasingly likely that they are infected with each return visit. We assume implicitly in the above that this cannot happen; however, we allow for the possibility of returning shoppers in the experiments given in the electronic supplementary material. Secondly, we remark that, in practice, $\xi$ will not be identical for all pairs of people in the shop. In particular, the compliance of the shoppers to guidelines may be difficult to achieve uniformly. Given the centrality of controlling $\xi$ to keeping the infection rate as low as possible, even a small minority of shoppers who do not comply with individual shop regulations could represent a significant and unnecessary risk for everyone present in the shop, as well as future users of the shop.

## 4. Conclusion

In this paper, we have examined the spread of COVID-19 in various shopping settings. As discussed in §2, considering the viral dose accumulated by the shoppers in the worst-case scenario is a question of critical importance. Looking only at the viral dose of susceptible individuals when shopping (rather than queuing), we find that efficiency of shopping is the largest driver of viral exposure. This is most important for shops with narrow aisles. At higher densities of people or more compact shops, the movement of shoppers can become inhibited. Supermarkets with better flowing layouts and a higher efficiency of shopping will be safer for individuals. One place where flowing movement is not necessarily possible in is within a queue.

However, queuing is often necessary in certain situations. For such cases, we find in §3 that a system with a cautious denominator population and well-protected members of staff can lead to very limited spread of COVID-19, while ensuring the shop remains economically viable. This appears to hold even under extremely pessimistic assumptions on the behaviour of the virus.

These results are necessarily based on incomplete information and many assumptions in the model about the physics and biology of the spread of COVID-19, and of the way that people go about their shopping. The principles which emerge should therefore be regarded as best estimates, to be updated when more data becomes available.

Data accessibility. This article has no additional data.

Competing interests. We declare we have no competing interests.

Funding. This work arose initially from the discussions at a Virtual Study Group (VSG) 29–30 April 2020, on *Guiding principles for unlocking the workforce*. The VSG was organized by the Virtual Forum for Knowledge Exchange in the Mathematical Sciences (V-KEMS). A fuller description of the overall results from the VSG are given in [1]. S.J. also acknowledges support from the Alan Turing Institute under EPSRC grant no. EP/N510129/1.

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
