## [Peer Review File · Royal Society Open Science]

Review History

RSOS-210344.R0 (Original submission)

Review form: Reviewer 1

Is the manuscript scientifically sound in its present form?

Yes

Are the interpretations and conclusions justified by the results?

Yes

Is the language acceptable?

Yes

Do you have any ethical concerns with this paper?

No

Have you any concerns about statistical analyses in this paper?

No

Recommendation?

Accept with minor revision (please list in comments)

Comments to the Author(s)

Review of Assessing Risk in the Retail Environment during the COVID-19 Pandemic.

This paper used mathematical modelling to investigate the rate of COVID-19 transmission during shopping (and visiting takeaways) and determine what strategies could minimise this probability e.g multiple queues and one way systems. Agent based models of shopper dynamics are used with COVID-19 transmission modelled by time integration to estimate an accumulated virion dose that is related to infection probability.

I note that the paper has already gone through a peer review process.

I recommend publication once a number of typos and minor issues which appear to be concentrated in section 2 are addressed

- P5 L31 f^{sd} missing, also be consistent f_{α} or f^{α} ?
- P6 L16 exp should be in roman not italics
- P6 L32 add brackets to make dimensions work
- P6 L55 units should be in roman not italics
- P7 Fig1 Label axes.
- P7 L58 Missing full stop.
- P8 L28 Variance has the wrong unit: standard deviation?
- P9 L19 Missing reference for matlab software?
- P10 L19 $\rho[r(t)]$ or $\rho(r,t)$? Be consistent. (I prefer the latter.)

P refers to manuscript page. L gives an approximate location in terms of the scale on the left hand side of the proof copy.

Review form: Reviewer 2

Is the manuscript scientifically sound in its present form?

No

Are the interpretations and conclusions justified by the results?

Yes

Is the language acceptable?

Yes

Do you have any ethical concerns with this paper?

No

Have you any concerns about statistical analyses in this paper?

No

Recommendation?

Accept with minor revision (please list in comments)

Comments to the Author(s)

In the supplementary materials, on the last line of page 3 following equation (3.1), the sentence should be about $P(\log_{10} \sigma) \neq P(\sigma)$.

Do you need Section 2(d) at all? At the moment it is not scientifically sound. To repeat, MID causes infection in 50% of cases - not the minimum needed for infection to occur (which your response to review seems to suggest). And you have no evidence to support any suggestion that $P(\cdot)$ (rather than $P(\log(\cdot))$) might be convex near the origin. If indeed the rest of the paper only considers the actual viral dose and how to minimise it then Section 2(d) is unnecessary.

Decision letter (RSOS-210344.R0)

Dear Dr Budd

On behalf of the Editors, we are pleased to inform you that your Manuscript RSOS-210344 "Assessing Risk in the Retail Environment during the COVID-19 Pandemic" has been accepted for publication in Royal Society Open Science subject to minor revision in accordance with the referees' reports. Please find the referees' comments along with any feedback from the Editors below my signature.

Please submit your revised manuscript and required files (see below) no later than 7 days from today's (ie 16-Apr-2021) date. Note: the ScholarOne system will 'lock' if submission of the revision is attempted 7 or more days after the deadline. If you do not think you will be able to meet this deadline please contact the editorial office immediately.

on behalf of Dr Jose Carrillo (Associate Editor) and Mark Chaplain (Subject Editor)
 openscience@royalsociety.org

Reviewer comments to Author:

Reviewer: 1

Comments to the Author(s)

Review of Assessing Risk in the Retail Environment during the COVID-19
 Pandemic.

This paper used mathematical modelling to investigate the rate of COVID-19 transmission during shopping (and visiting takeaways) and determine what strategies could minimise this probability e.g multiple queues and one way systems. Agent based models of shopper dynamics are used with COVID-19 transmission modelled by time integration to estimate an accumulated virion dose that is related to infection probability.

I note that the paper has already gone through a peer review process.

I recommend publication once a number of typos and minor issues which appear to be concentrated in section 2 are addressed

- P5 L31 f^{sd} missing, also be consistent f_{α} or f^{α} ?
- P6 L16 exp should be in roman not italics
- P6 L32 add brackets to make dimensions work
- P6 L55 units should be in roman not italics
- P7 Fig1 Label axes.
- P7 L58 Missing full stop.
- P8 L28 Variance has the wrong unit: standard deviation?
- P9 L19 Missing reference for matlab software?
- P10 L19 $\rho[r(t)]$ or $\rho(r,t)$? Be consistent. (I prefer the latter.)

P refers to manuscript page. L gives an approximate location in terms of the scale on the left hand side of the proof copy.

Reviewer: 2

Comments to the Author(s)

In the supplementary materials, on the last line of page 3 following equation (3.1), the sentence should be about $P(\log_{10} \sigma) \text{ not } P(\sigma)$.

Do you need Section 2(d) at all? At the moment it is not scientifically sound. To repeat, MID causes infection in 50% of cases - not the minimum needed for infection to occur (which your response to review seems to suggest). And you have no evidence to support any suggestion that $P(\cdot)$ (rather than $P(\log(\cdot))$) might be convex near the origin. If indeed the rest of the paper only considers the actual viral dose and how to minimise it then Section 2(d) is unnecessary.

===PREPARING YOUR MANUSCRIPT===

===PREPARING YOUR REVISION IN SCHOLARONE===

- If you are providing image files for potential cover images, please upload these at this step, and inform the editorial office you have done so. You must hold the copyright to any image provided.
- A copy of your point-by-point response to referees and Editors. This will expedite the preparation of your proof.

- Ensure that your data access statement meets the requirements at <https://royalsociety.org/journals/authors/author-guidelines/#data>. You should ensure that you cite the dataset in your reference list. If you have deposited data etc in the Dryad repository, please only include the 'For publication' link at this stage. You should remove the 'For review' link.
- If you are requesting an article processing charge waiver, you must select the relevant waiver option (if requesting a discretionary waiver, the form should have been uploaded at Step 3 'File upload' above).
- If you have uploaded ESM files, please ensure you follow the guidance at <https://royalsociety.org/journals/authors/author-guidelines/#supplementary-material> to include a suitable title and informative caption. An example of appropriate titling and captioning may be found at https://figshare.com/articles/Table_S2_from_Is_there_a_trade-off_between_peak_performance_and_performance_breadth_across_temperatures_for_aerobic_scope_in_teleost_fishes_/3843624.

Author's Response to Decision Letter for (RSOS-210344.R0)

See Appendix A.

Decision letter (RSOS-210344.R1)

Dear Dr Budd,

It is a pleasure to accept your manuscript entitled "Assessing Risk in the Retail Environment during the COVID-19 Pandemic" in its current form for publication in Royal Society Open Science.

COVID-19 rapid publication process:

We are taking steps to expedite the publication of research relevant to the pandemic. If you wish, you can opt to have your paper published as soon as it is ready, rather than waiting for it to be published the scheduled Wednesday.

This means your paper will not be included in the weekly media round-up which the Society sends to journalists ahead of publication. However, it will still appear in the COVID-19 Publishing Collection which journalists will be directed to each week (<https://royalsocietypublishing.org/topic/special-collections/novel-coronavirus-outbreak>).

If you wish to have your paper considered for immediate publication, or to discuss further, please notify openscience_proofs@royalsociety.org and press@royalsociety.org when you respond to this email.

on behalf of Dr Jose Carrillo (Associate Editor) and Mark Chaplain (Subject Editor)
openscience@royalsociety.org

Appendix A

Assessing Risk in the Retail Environment during the COVID-19 Pandemic

Response to Reviewers

The authors would like to take this opportunity to thank the reviewers for their quick and constructive feedback on this version of the paper.

Reviewer 1

This paper used mathematical modelling to investigate the rate of COVID-19 transmission during shopping (and visiting takeaways) and determine what strategies could minimise this probability e.g multiple queues and one way systems. Agent based models of shopper dynamics are used with COVID-19 transmission modelled by time integration to estimate an accumulated virion dose that is related to infection probability.

I note that the paper has already gone through a peer review process.

I recommend publication once a number of typos and minor issues which appear to be concentrated in section 2 are addressed

- P5 L31 f^{α} missing, also be consistent f_{α} or f^{α} ?
- P6 L16 exp should be in roman not italics
- P6 L32 add brackets to make dimensions work
- P6 L55 units should be in roman not italics
- P7 Fig1 Label axes.
- P7 L58 Missing full stop.
- P8 L28 Variance has the wrong unit: standard deviation?
- P9 L19 Missing reference for matlab software?
- P10 L19 $\rho[r(t)]$ or $\rho(r,t)$? Be consistent. (I prefer the latter.)

P refers to manuscript page. L gives an approximate location in terms of the scale on the left hand side of the proof copy.

Many thanks for pointing out these typos in Section 2. We have now corrected all of them.

Reviewer 2

In the supplementary materials, on the last line of page 3 following equation (3.1), the sentence should be about $P(\log_{10} \sigma) \neq P(\sigma)$.

Thank you for spotting this - we have now removed this subsection from the supps as we have also removed Section 2(d) from the main document (see below).

Do you need Section 2(d) at all? At the moment it is not scientifically sound. To repeat, MID causes infection in 50% of cases - not the minimum needed for infection to occur (which your response to review seems to suggest). And you have no evidence to support any suggestion that $P(\cdot)$ (rather than $P(\log(\cdot))$) might be convex near the origin. If indeed the rest of the paper only considers the actual viral dose and how to minimise it then Section 2(d) is unnecessary.

Many thanks for this point - we have now removed Section 2(d) from the Main paper, and the related section 3 from the Supplementary materials.